# Enhancing Contrastive Learning with Variable Similarity

**Haowen Cui**[1], **Shuo Chen**[2,3]*, **Jun Li**[1], **Jian Yang**[1]*
[1]PCA Lab,† Nanjing University of Science and Technology, China
[2]School of Intelligence Science and Technology, Nanjing University, China
[3]Center for Advanced Intelligence Project, RIKEN National Science Institute, Japan

## Abstract

Contrastive learning has achieved remarkable success in self-supervised learning by pretraining a generalizable feature representation based on the augmentation invariance. Most existing approaches assume that different augmented views of the same instance (i.e., the *positive pairs*) remain semantically invariant. However, the augmentation results with *varying extent* may introduce semantic discrepancies or even content distortion, and thus the conventional (pseudo) supervision from augmentation invariance may lead to misguided learning objectives. In this paper, we propose a novel method called Contrastive Learning with Variable Similarity (CLVS) to accurately characterize the intrinsic similarity relationships between different augmented views. Our method dynamically adjusts the similarity based on the augmentation extent, and it ensures that strongly augmented views are always assigned lower similarity scores than weakly augmented ones. We provide a theoretical analysis to guarantee the effectiveness of the variable similarity in improving model generalizability. Extensive experiments demonstrate the superiority of our approach, achieving gains of 2.1% on ImageNet-100 and 1.4% on ImageNet-1k compared with the state-of-the-art methods.

## 1 Introduction

Learning effective feature representations [1, 2, 3] is a fundamental task in machine learning, with profound implications for various applications, including image classification [4, 5], object detection [6, 7], and segmentation [8, 9]. In recent years, self-supervised learning has emerged as a leading paradigm for unsupervised visual representation learning [10, 11, 12, 13]. Among various pretext tasks in self-supervised learning, contrastive learning constructs self-supervisory signals by treating different augmented views of the same image as positive pairs, enabling the extraction of high-quality feature representations without extensive labeled data. This paradigm has achieved remarkable success, demonstrating its potential to bridge the gap between supervised and unsupervised learning.

---

*Corresponding Authors
†PCA Lab, Key Lab of Intelligent Perception and Systems for High-Dimensional Information of Ministry of Education, School of Computer Science and Engineering, Nanjing University of Science and Technology

39th Conference on Neural Information Processing Systems (NeurIPS 2025).

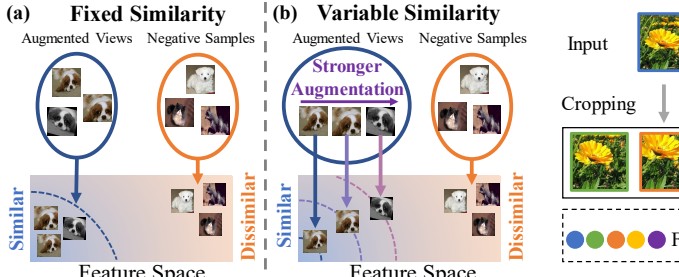

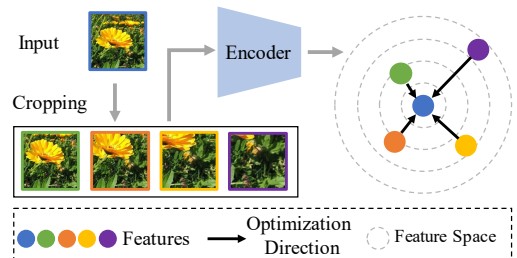

Figure 1: Illustration of different similarity measure in contrastive learning. (a) Fixed maximum similarity among augmented views of the same instance. (b) Variable similarity (ours) among augmented views of the same instance.

Figure 2: Example of learning framework with the cropping augmentation in contrastive learning. Despite the semantic inconsistency between different augmented views, current contrastive learning method equally optimize them towards the direction with the maximum similarity.

As illustrated in Fig. 1(a), the core mechanism of contrastive learning is to maximize the similarity of embeddings between different views generated by data augmentations. These augmented views share the same underlying semantic concepts, enabling the model to learn invariant representations and thereby improving its generalization ability [10, 14, 15]. However, the assumption that all augmented views are equally valid representations of the original data is still flawed, as the semantic consistency between these views cannot always be guaranteed. As shown in Fig. 2, over-cropping augmentation can produce views containing only background, leading to significant semantic discrepancies between these views and others. This inconsistency may force the model to optimize in the wrong direction, compromising its ability to learn meaningful representations.

In fact, data augmentations inherently induce a spectrum of semantic consistency between views, rather than a binary state of similar or dissimilar. As shown in Fig. 2, the progressive loss of critical features becomes more pronounced as the extent of cropping augmentation increases. This observation suggests that the similarity usually varies dynamically, rather than remaining fixed. Inspired by the above observation, we argue that the similarity between augmented views in contrastive learning should be *variable* according to the extent of applied data augmentations, as illustrated in Fig. 1(b). This variability allows the model to reflect the semantic variation introduced by data augmentations. In particular, the stronger data augmentation leads to more significant information degradation, resulting in a lower similarity compared with weakly augmented views.

In this paper, we propose Contrastive Learning with Variable Similarity (CLVS), which models the similarity between views as a variable determined by their augmentation parameters. Specifically, we first build an augmentation-aware module to predict the variable similarity between two augmented views based on their augmentation parameters. Then, we introduce an alignment objective that constrains the similarity between augmented views to align with the predicted variable similarity, thereby guiding the representations to reflect varying semantic relations. Additionally, the loss function will penalize cases when the similarity of strongly augmented views exceeds that of weakly augmented views. To further validate our approach, we provide a theoretical analysis demonstrating that the generalization error bound of our method can be effectively shrunk through the use of variable similarity. Our method is a general-purpose technique that can be easily integrated into many existing contrastive learning frameworks to enhance their performance. When applied to MoCo [16] and SimSiam [12], our approach achieves significant improvements of 6.1% and 5.9% on the ImageNet-100 dataset, demonstrating its strong generalizability and practical effectiveness.

In summary, our main contributions are as follows:

- We enhance contrastive learning by introducing the variable similarity to capture variations between augmented views, supported by an augmentation-aware module that estimates accurate semantic similarity between these views based on their augmentation parameters.

- We theoretically prove that the statistical variance of similarity decreases with the utilization of variable similarity, thereby resulting in a reduction in the error bound of the generalization.

- Experimental results conducted on standard benchmarks demonstrate the superiority of our method, surpassing the state-of-the-art methods by 2.1% on ImageNet-100 and 1.4% on ImageNet-1k, respectively.

## 2 Related Works

### 2.1 Contrastive Learning

Contrastive learning has recently emerged as a leading paradigm in self-supervised learning [10, 11, 12, 13, 17, 18, 19], gradually closing the performance gap with supervised learning. The core principle of contrastive learning is to pull together the embedding features of positive pairs, which are different augmented views of the same instance. This aligns with the broader objective of metric learning [20, 21, 22, 23, 24, 25], where the goal is to learn representations that preserve semantic similarity in the embedding space. A crucial challenge in contrastive learning is to prevent the representation collapse, where all samples are mapped to the same representation, resulting in a loss of discriminative capability. Several negative-used contrastive methods [10, 11, 26, 16, 27] tackle this issue using the InfoNCE criterion [28]. This criterion pulls positive pairs together while pushing negative pairs apart. Negative-free methods [13, 12, 29] mitigate the collapse problem by leveraging only positive pairs and introducing an asymmetric design, i.e., the stop-gradient mechanism [30]. Additionally, some works [17, 18, 31, 32] aim to alleviate the collapse problem by maximizing the information content.

### 2.2 Augmentation Technique

Data augmentation plays a crucial role in many representation learning tasks. It improves the quality and diversity of positive sample pairs, enhances the robustness of learned representations [14, 33], and can help mitigate collapse [34]. Recent studies have explored various strategies to leverage data augmentation more effectively. For instance, AugSelf [35] preserves augmentation-aware information through an auxiliary model that predicts the difference between augmentation parameters of augmented views. HAIEV [36] argues that different augmentations should be treated unequally and proposes computing contrastive loss at different network layers to achieve a broader distribution of augmentation invariance. LoGo [37] simultaneously incorporates global and local views, encouraging local views to have distinct representations. EquiMod [38] introduces equivariance into contrastive learning, aiming to better capture the augmentation information and improve the robustness of learned representations. Other works focus on leveraging stronger augmentations: CLSA [39] aims to minimize the distribution divergence between weak and strong augmented views, while RényiCL [40] generalizes representation learning by utilizing Rényi divergence as the learning objective. CoCor [41] focuses on exploiting the strength of the composite data augmentation quantified by frequency, but it does not explicitly model the augmentation parameters. As a result, it cannot ensure the variation within the same augmentation type or ensure that the similarity between augmented views aligns with the extent of augmentation.

While these methods typically treat all positive pairs as equally similar, they usually ignore the fact that different augmented views may preserve varying levels of semantic content. To address this limitation, we propose the concept of variable similarity, which better captures the nuanced relationships between views with diverse augmentations.

## 3 Method

### 3.1 Preliminaries of Contrastive Learning

Contrastive learning aims to train a generalizable feature encoder $\boldsymbol{f} : \mathbb{R}^n \rightarrow \mathbb{R}^d$ that maps input data from the original $n$-dimensional space to a $d$-dimensional feature space. Given an input instance $\boldsymbol{x} \in \mathbb{R}^n$ sampled from the dataset $\mathbb{D}$ and a distribution of data augmentations $\mathcal{T}$, the training objective is to maximize the similarity between the embeddings $\boldsymbol{z}_1 = \boldsymbol{f}(t_1(\boldsymbol{x}))$ and $\boldsymbol{z}_2 = \boldsymbol{f}(t_2(\boldsymbol{x}))$ of two augmented views $t_1(\boldsymbol{x})$ and $t_2(\boldsymbol{x})$, where $t_1, t_2 \sim \mathcal{T}$. This can be formulated as minimizing the following general objective:

$$\mathcal{L}_{con}(\boldsymbol{f}) = \mathbb{E}_{x \sim \mathbb{D}, t_{1,2} \sim \mathcal{T}}[\ell(\boldsymbol{z}_1, \boldsymbol{z}_2)], \tag{1}$$

where $\ell(\cdot, \cdot)$ is an empirical loss to evaluate the inconsistency between $\boldsymbol{z}_1$ and $\boldsymbol{z}_2$. For the traditional contrastive methods [10, 11], the InfoNCE criterion [28] is typically used as the loss function:

$$\ell(\boldsymbol{z}_1, \boldsymbol{z}_2) = -log \frac{\mathrm{e}^{sim(\boldsymbol{z}_1 \cdot \boldsymbol{z}_2)/\tau}}{\sum_i^N \mathrm{e}^{sim(\boldsymbol{z}_1 \cdot \boldsymbol{z}_i)/\tau}}, \tag{2}$$

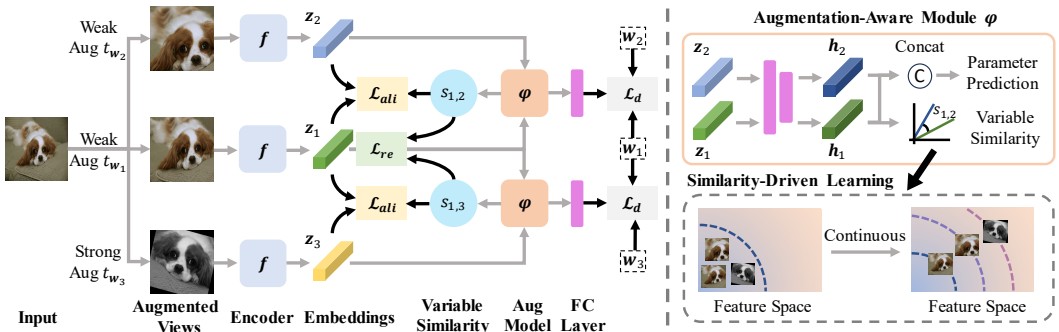

Figure 3: Overview of the CLVS. The encoder $f$ extracts the embedding $z_i$ of augmented views. The augmentation-aware module $\varphi$ predicts the variable similarity between two augmented views. The similarity between the embeddings of two augmented views is constrained by $\mathcal{L}_{ali}$ to approximate the predicted variable similarity during training. $\mathcal{L}_{re}$ enforces the similarity to the weakly augmented view $s_{1,2}$ larger than that to the strongly augmented view $s_{1,3}$. By incorporating $\mathcal{L}_d$, the augmentation-aware module $\varphi$ gains the ability to perceive augmentations.

where $sim(\cdot, \cdot)$ is the similarity measurement function, such as cosine similarity. $\tau$ is a temperature hyperparameter and $N$ is the number of negative samples in a mini-batch. This encourages the model to pull positive pairs closer while pushing negative pairs apart in the embedding space.

### 3.2 Variable Similarity in Contrastive Learning

To address the limitation of enforcing a fixed similarity for positive pairs in contrastive learning, we propose CLVS. Unlike traditional methods, CLVS dynamically adjusts the similarity between augmented views based on the parameters of applied augmentations. This dynamic adjustment enables the model to better capture semantic relationships and produce more robust representations.

**Augmentation-Aware Learning.** Our key idea is to make the similarity between augmented views of an input instance $x$ consistent with the extent of their augmentations. Let $w_i = (w_i^1, ..., w_i^c)$ denote the augmentation parameters for transformation $t_{w_i}$, where the vector $w_i^j$ represents the parameter of the $j$-th augmentation type (e.g., cropping, color jittering), and $c$ is the number of augmentation types. Inspired by [35], we introduce an auxiliary task to predict the differences in augmentation parameters. Specifically, for the $j$-th augmentation type, the difference between $t_{w_1}$ and $t_{w_2}$ is defined as $d_{1,2}^j = w_1^j - w_2^j$. To make the learned representations sensitive to these differences, we build an augmentation-aware module $\varphi = (\varphi^1, ..., \varphi^c)$, where each $\varphi^j : \mathbb{R}^d \to \mathbb{R}^m$ is implemented as a Multi-Layer Perceptron (MLP) for generating augmentation-aware embeddings. Here, $m$ is the embedding dimensionality. Each $\varphi^j$ projects the feature embeddings $z_1$ and $z_2$ (produced by the encoder $f$) into an augmentation-aware latent space, yielding $h_1^j = \varphi^j(z_1)$ and $h_2^j = \varphi^j(z_2)$. The concatenated embedding of $h_1^j$ and $h_2^j$ is then fed into a predictor $\psi^j$ to estimate the parameter difference $\hat{d}_{1,2}^j$. This prediction task encourages the features to retain augmentation-aware information, allowing them to better reflect the extent of augmentations, and thereby laying the foundation for variable similarity estimation. The learning process for this auxiliary task is summarized by:

$$\mathcal{L}_d(f, \varphi, \psi) = \sum_{j=1}^c \ell_j(\hat{d}_{1,2}^j, d_{1,2}^j) = \sum_{j=1}^c \ell_j(\psi^j(\text{concat}(\varphi^j(z_1), \varphi^j(z_2))), d_{1,2}^j). \quad (3)$$

The loss function $\ell_j$ dynamically adjusts depending on the type of data augmentation. [3]

**Variable Similarity Alignment.** Building on the above auxiliary task, we calculate the variable similarity through an adaptive selection mechanism governed by $\varphi$. In this paper, we use $\hat{s}_{i,j}$ to denote the predicted variable similarity, and $s_{i,j}$ for the similarity between encoder outputs of two augmented views. For each augmentation type $j$, we calculate the similarity between $h_1^j$ and $h_2^j$ as:

---

[3]For details, $\ell_j = ||\hat{d}_{1,2}^j - d_{1,2}^j||_2$ for augmentations with parameters, e.g., cropping, color jittering. $\ell_j = d_{1,2}^j \log(\hat{d}_{1,2}^j) + (1 - d_{1,2}^j) \log(1 - \hat{d}_{1,2}^j)$ for augmentations without parameters, e.g., flipping, grayscaling.

$\hat{s}_{1,2}^j = sim(\boldsymbol{h}_1^j, \boldsymbol{h}_2^j) = \boldsymbol{\varphi}^j(\boldsymbol{z}_1) \cdot \boldsymbol{\varphi}^j(\boldsymbol{z}_2)/(||\boldsymbol{\varphi}^j(\boldsymbol{z}_1)||_2||\boldsymbol{\varphi}^j(\boldsymbol{z}_2)||_2)$, i.e., $sim(\cdot, \cdot)$ represents cosine similarity, where $||\cdot||_2$ denotes the $\ell_2$ normalization. This is motivated by the intuition that smaller augmentation parameter differences indicate more consistent transformations, and thus a higher semantic similarity. Since different types of augmentation have varying impacts on the similarity, we aggregate the similarities across all augmentation types. To ensure that the variable similarity captures the most significant discrepancies between views, we calculate the minimum similarity across all augmentation types. The variable similarity between two augmented views is defined as:

$$\hat{s}_{1,2} = \min_{j \in \{1,...,c\}} sim(\boldsymbol{\varphi}^j(\boldsymbol{z}_1), \boldsymbol{\varphi}^j(\boldsymbol{z}_2)). \tag{4}$$

To integrate the above predicted variable similarity into the contrastive learning framework, we align the cosine similarity between the embeddings of two augmented views with the predicted variable similarity via an $\ell_2$ loss:

$$\mathcal{L}_{ali}(\boldsymbol{f}, \boldsymbol{\varphi}) = \mathbb{E}_{\boldsymbol{x}, t_{\boldsymbol{w}_1, \boldsymbol{w}_2}}[\ell_{ali}(\hat{s}_{1,2}, s_{1,2})] = \mathbb{E}_{\boldsymbol{x}, t_{\boldsymbol{w}_1, \boldsymbol{w}_2}}\left[\left\|\min_{j \in \{1,...,c\}} sim(\boldsymbol{\varphi}^j(\boldsymbol{z}_1), \boldsymbol{\varphi}^j(\boldsymbol{z}_2)) - sim(\boldsymbol{z}_1, \boldsymbol{z}_2)\right\|_2^2\right]. \tag{5}$$

By minimizing $\mathcal{L}_{ali}$, the model learns to adaptively adjust the similarity objective for each positive pair based on the predicted variable similarity, addressing semantic inconsistencies caused by data augmentations.

**Relative Similarity Constraint.** To ensure the validity of the predicted variable similarity, we introduce a constraint based on the observation that stronger augmentations cause greater information degradation, thereby reducing the semantic consistency between augmented views. Specifically, the similarity between a view and its strongly augmented version should be lower than its similarity to a weakly augmented view. Given a stronger augmentation $t_{w_3}$, the relative similarity constraint can be formalized as follows:

$$\begin{aligned} \mathcal{L}_{re}(\boldsymbol{f}, \boldsymbol{\varphi}) &= \mathbb{E}_{x, t_{w_1, w_2, w_3}}[\ell_{con}(\hat{s}_{1,2}, \hat{s}_{1,3})] \\ &= \mathbb{E}_{x, t_{w_1, w_2, w_3}}\{\max[0, \min_{j \in 1,...,c}(sim(\boldsymbol{\varphi}^j(\boldsymbol{z}_1), \boldsymbol{\varphi}^j(\boldsymbol{z}_3))) - \min_{j \in 1,...,c}(sim(\boldsymbol{\varphi}^j(\boldsymbol{z}_1), \boldsymbol{\varphi}^j(\boldsymbol{z}_2)))]\}. \end{aligned} \tag{6}$$

Here, $\hat{s}_{1,3}$ represents the predicted variable similarity between the weakly augmented view $t_{w_1}(x)$ and the strongly augmented view $t_{w_3}(x)$. The max operator in Eq. (6) ensures that the loss is only activated when the similarity to a strongly augmented view exceeds that to a weakly augmented view, thus enforcing the desired relationship between augmentation strength and semantic consistency.

The overview of CLVS is shown in Fig. 3. The total training loss consists of four components: the base contrastive loss, the parameter prediction loss, the alignment loss for variable similarity, and the constraint loss for augmentation consistency. Formally, the total loss is defined as:

$$\mathcal{L}_{total} = \mathcal{L}_{con}(\boldsymbol{f}) + \omega\mathcal{L}_d(\boldsymbol{f}, \boldsymbol{\varphi}, \boldsymbol{\psi}) + \lambda\mathcal{L}_{ali}(\boldsymbol{f}, \boldsymbol{\varphi}) + \gamma\mathcal{L}_{re}(\boldsymbol{f}, \boldsymbol{\varphi}), \tag{7}$$

where $\omega$, $\lambda$, and $\gamma$ are positive weight coefficients that balance the contributions of different loss terms. By combining these losses, our method preserves the core objective of contrastive learning while addressing the limitation caused by information degradation during data augmentation, resulting in more robust and semantically meaningful representations.

### 3.3 Theoretical Analysis of Variable Similarity

In this section, we aim to demonstrate that our proposed learning algorithm enhances the Generalization Error Bound (GEB) [42] compared to traditional contrastive learning methods. The GEB typically characterizes how well a model trained on empirical data performs on unseen samples. For self-supervised learning, although the learned encoders are used in different recognition tasks, such an error bound of the learning objective can still provide a quantitative result to evaluate the reliability of the encoder on unseen test data. This is because that the lower generalization error is expected to bring about the smaller InfoNCE loss on the test data, and thus the corresponding feature discriminability during test phase can be ensured. Here, we establish that the proposed new loss terms $\mathcal{L}_d$, $\mathcal{L}_{ali}$, and $\mathcal{L}_{re}$ contribute to tightening the GEB, thereby validating the efficacy of our method.

To provide a more detailed understanding, we consider the underlying data distribution $\mathscr{D}$, and introduce the expected risk, defined as $\widehat{\mathcal{L}}_{total}(\boldsymbol{f}, \boldsymbol{\varphi}; \mathscr{D}) = \mathbb{E}_{\{\boldsymbol{t}_i|\boldsymbol{t}_i \sim \mathscr{D}\}_{i=1}^N}[\mathcal{L}_{total}(\boldsymbol{f}, \boldsymbol{\varphi}; \{\boldsymbol{t}_i\}_{i=1}^N)]$. The expected risk represents the true objective we aim to minimize, while the empirical risk $\mathcal{L}_{total}(\boldsymbol{f}, \boldsymbol{\varphi}; \mathscr{D})$

Table 1: Top-1 accuracies of linear evaluation (%). All compared methods with a ResNet-50 encoder are pretrained for 200 epochs on the ImageNet-100 dataset.

| Method | Batch Size | Top-1 |
|---|---|---|
| MoCo [16] | 256 | 78.8 |
| SimSiam [12] | 256 | 79.1 |
| AugSelf [35] | 256 | 80.5 |
| RényiCL [40] | 256 | 82.1 |
| EquiMod [38] | 256 | 83.9 |
| CoCor [41] | 256 | 83.7 |
| GCA [44] | 256 | 72.4 |
| INTL [32] | 256 | 83.9 |
| **MoCo+CLVS** | 256 | **84.9** (+6.1) |
| **SimSiam+CLVS** | 256 | **86.0** (+5.9) |

Table 2: Top-1 accuracies of linear evaluation (%). All compared methods with a ResNet-50 encoder are pretrained for 200 epochs on the ImageNet-1k dataset.

| Method | Batch Size | Top-1 |
|---|---|---|
| SimCLR [10] | 4096 | 68.3 |
| MoCo [16] | 256 | 67.5 |
| SimSiam [12] | 256 | 70.0 |
| AugSelf [35] | 256 | 69.4 |
| SwAV [26] | 4096 | 69.1 |
| EquiMod [38] | 256 | 67.6 |
| GCA [44] | 256 | 56.1 |
| INTL [32] | 256 | 69.9 |
| **MoCo+CLVS** | 256 | **69.9** (+2.4) |
| **SimSiam+CLVS** | 256 | **71.4** (+1.4) |

Table 3: Linear evaluation accuracies (%) on various datasets. All compared methods with a ResNet-50 encoder are pretrained for 200 epochs on the ImageNet-100 dataset.

| Method | CIFAR10 | CIFAR100 | Caltech101 | SUN397 | Food | Flowers | Pets |
|---|---|---|---|---|---|---|---|
| MoCo [16] | 85.81 | 64.96 | 85.78 | 48.73 | 62.37 | 84.01 | 69.15 |
| SimSiam [12] | 86.52 | 65.98 | 87.21 | 49.68 | 62.67 | 84.09 | 69.28 |
| AugSelf [35] | 88.10 | 68.48 | 88.95 | 50.65 | 65.56 | 88.60 | 71.93 |
| RényiCL [40] | 86.60 | 64.92 | 87.21 | 48.45 | 61.89 | 85.33 | 74.87 |
| EquiMod [38] | 87.86 | 69.59 | 90.37 | 52.85 | 65.71 | 89.88 | 75.88 |
| CoCor [41] | 86.89 | 66.36 | 87.85 | 49.92 | 62.48 | 87.95 | 76.04 |
| GCA [44] | 76.69 | 51.87 | 72.24 | 37.20 | 44.96 | 68.21 | 49.55 |
| INTL [32] | 87.55 | 67.79 | 88.77 | 51.62 | 63.05 | 87.75 | 75.22 |
| **MoCo+CLVS** | 87.29 (+1.48) | 67.22 (+2.26) | 88.93 (+3.15) | 52.55 (+3.82) | 65.07 (+2.70) | 87.48 (+3.47) | **77.65** (+8.50) |
| **SimSiam+CLVS** | **89.92** (+3.40) | **71.37** (+5.42) | **91.73** (+4.52) | **54.80** (+5.12) | **68.97** (+6.30) | **91.12** (+7.03) | 77.16 (+7.88) |

is computed from a finite dataset. The gap between these two quantities, known as the generalization error, reflects the model's ability to generalize beyond the training set.

**Theorem 1.** *For any* $\{f, \varphi\}$ *learned from the objective* $\mathcal{L}_{total}(f, \varphi)$ *and any given constant* $\delta \in (0, 1)$, *we have that with probability at least* $1 - \delta$,

$$|\mathcal{L}_{total}(f, \varphi; \mathscr{D}) - \widehat{\mathcal{L}}_{total}(f, \varphi; \mathscr{D})| \leq C_1/(C_2 + \omega)\log(1 + \rho(\lambda, \gamma))\sqrt{[\ln(2/\delta)]/(2N)}, \qquad (8)$$

*where* $\rho(\lambda, \gamma) = \sum_{i<j}(s_{ij} - \bar{s})/C_N^2 > 0$ *is monotonically decreasing w.r.t.* $\lambda$ *and* $\gamma$. *Here* $C_1/(C_2 + \omega)$ *is monotonically decreasing w.r.t.* $\omega$, *where* $C_1, C_2 > 0$ *are constants that independent of* $f$ *and* $\varphi$.

From the result presented in the existing work [43], which highlights the relationship between larger data volumes and improved model generalization. This decrease reflects the statistical principle that larger datasets provide more reliable approximations of the underlying data distribution, thereby reducing the generalization gap. Meanwhile, we observe that the error bound becomes tighter as the parameter $\omega$ increases. This is because that the augmentation-specific loss effectively constrains the consistency between similarities, thereby enhancing the model generalization ability. More importantly, we can find that such an error bound becomes *tighter* as $\lambda$ and $\gamma$ increase. Actually, this is due to the further decrease of the variance $\rho(\lambda, \gamma) = \sum_{i<j}(s_{ij} - \bar{s})/C_N^2$ induced by the parameters $\lambda$ and $\gamma$. Intuitively, since the similarity values are constrained to be floats between 0 and 1 rather than those extreme values of either 0 or 1, they are more likely located in the central area of $[0, 1]$, and thus the variance is naturally reduced. Notably, our theoretical analysis is task-agnostic and focuses on the generalization behavior of the training objective itself, rather than any specific downstream application.

In summary, the above theorem demonstrates that our method successfully improves the generalization performance of conventional contrastive learning algorithms by leveraging both increased data and the new loss terms.

Table 4: Transfer learning results on VOC and COCO object detection tasks. All compared methods with a ResNet-50 encoder are pretrained for 200 epochs on the ImageNet-100 dataset.

| Method | VOC07+12 | | | COCO | | | | | |
|---|---|---|---|---|---|---|---|---|---|
| | AP | $AP_{50}$ | $AP_{75}$ | AP | $AP_{50}$ | $AP_{75}$ | $AP_s$ | $AP_m$ | $AP_l$ |
| MoCo [16] | 40.87 | 69.78 | 41.41 | 37.29 | 56.36 | 40.14 | 20.63 | 41.63 | 51.54 |
| SimSiam [12] | 39.83 | 67.27 | 40.09 | 37.24 | 56.61 | 40.25 | 21.21 | 41.78 | 50.26 |
| Augself [35] | 46.96 | 73.88 | 50.27 | 37.91 | 57.10 | 41.03 | 21.58 | 42.44 | 51.36 |
| RényiCL [40] | 41.59 | 72.76 | 41.63 | 37.82 | 57.16 | 40.46 | 21.90 | 42.37 | 51.31 |
| EquiMod [38] | 32.24 | 63.17 | 28.05 | 35.30 | 54.95 | 37.82 | 18.11 | 39.65 | 49.03 |
| CoCor [41] | **54.31** | **80.60** | **59.58** | 34.80 | 56.29 | 37.05 | 17.53 | 39.88 | 51.74 |
| GCA [44] | 30.80 | 60.80 | 27.39 | 34.93 | 54.04 | 37.17 | 20.09 | 39.05 | 48.34 |
| INTL [32] | 50.47 | 78.64 | 54.95 | 38.14 | 57.05 | 41.07 | 20.46 | 42.00 | 52.01 |
| **MoCo+CLVS** | 52.22 (+11.35) | 79.43 (+9.65) | 56.87 (+15.46) | **39.35** (+2.06) | **58.90** (+2.54) | **42.91** (+2.77) | **22.06** (+1.43) | **44.02** (+2.39) | **54.75** (+3.21) |
| **SimSiam+CLVS** | 53.20 (+13.37) | 79.13 (+11.86) | 58.39 (+18.30) | 38.24 (+1.00) | 57.51 (+0.90) | 41.37 (+1.12) | 21.71 (+0.50) | 42.58 (+0.80) | 52.52 (+2.26) |

Table 5: Few-shot classification accuracy with 95% confidenceinterval averaged over 2000 episodes on FC100, CUB200 and Plant Disease. $(N, K)$ denotes N-way K-shot task. All compared methods with a ResNet-50 encoder are pretrained for 200 epochs on the ImageNet-100 dataset.

| Method | FC100 | | CUB200 | | Plant | |
|---|---|---|---|---|---|---|
| | $(5, 1)$ | $(5, 5)$ | $(5, 1)$ | $(5, 5)$ | $(5, 1)$ | $(5, 5)$ |
| MoCo [16] | 38.84±0.39 | 54.53±0.38 | 41.63±0.46 | 55.96±0.46 | 68.42±0.49 | 86.22±0.34 |
| SimSiam [12] | 36.04±0.37 | 53.83±0.39 | 39.31±0.43 | 55.14±0.47 | 71.00±0.50 | 88.39±0.33 |
| Augself [35] | 40.73±0.41 | 58.41±0.39 | 42.50±0.44 | 59.97±0.45 | 73.66±0.49 | 90.23±0.31 |
| RényiCL [40] | 37.79±0.36 | 55.75±0.39 | 42.30±0.47 | 58.79±0.46 | 73.51±0.46 | 89.67±0.31 |
| EquiMod [38] | 44.83±0.41 | 62.76±0.40 | 44.81±0.46 | 62.92±0.46 | 77.21±0.45 | 91.16±0.25 |
| GCA [44] | 30.34±0.33 | 44.68±0.37 | 37.61±0.41 | 52.68±0.45 | 64.63±0.51 | 86.03±0.33 |
| INTL [32] | 44.96±0.40 | 62.49±0.40 | 44.79±0.48 | 62.27±0.48 | 77.05±0.46 | 91.92±0.28 |
| **MoCo+CLVS** | 40.28±0.40 (+1.44) | 56.26±0.39 (+1.73) | 42.77±0.47 (+1.14) | 57.98±0.46 (+2.02) | 71.16±0.49 (+2.74) | 88.54±0.32 (+2.32) |
| **SimSiam+CLVS** | **45.30±0.42** (+9.26) | **62.97±0.39** (+9.14) | **47.23±0.48** (+7.92) | **63.57±0.47** (+8.43) | **78.70±0.45** (+7.70) | **92.11±0.28** (+3.72) |

## 4 Experiments

In this section, we first present the implementation details of the experiments. Then, we perform extensive experiments on downstream tasks and compare our method with existing state-of-the-art methods. Finally, we provide ablation studies of the proposed method.

### 4.1 Implementation Details

We pretrain the standard ResNet-50 [4] encoder on the ImageNet-100 and ImageNet-1k [45] datasets. The encoder is pretrained for 200 epochs with the batch size of 256. Each model in $\varphi$ is implemented with 2 fully connected layers, and each predictor in $\psi$ consists of a single fully connected layer. We set $\omega = 0.5$ following [35], $\lambda = 0.5$, and $\gamma = 1$ in the training loss, respectively. To evaluate the effectiveness of CLVS in modeling variable similarity between positive samples, we adopt both the negative-used contrastive method MoCo [16] and the negative-free method SimSiam [12] as baselines.

For weak augmentations, we employ standard augmentation strategies [12], including *random cropping*, *color jittering*, *horizontal flipping*, *grayscale conversion*, and *gaussian blurring*. For strong augmentations, we integrate *RandAugment* [46] into the weak augmentation, which automates the selection of augmentation types and their magnitudes. The parameters of augmentations are set according to [35].

### 4.2 Main Results

In this subsection, we present experimental results on various downstream tasks. We compare CLVS with state-of-the-art contrastive learning methods. For clarity, the best results are highlighted in **bold**, while the second-best results are underlined.

**Comparison on Linear Evaluation.** We initially pretrain CLVS and perform linear evaluation on the ImageNet-100 and ImageNet-1k datasets. The linear evaluation protocol follows the approach of [12]. The experimental results are shown in Tab. 1 and Tab. 2, respectively. Our proposed method achieves

Table 6: Comparison with methods using stronger augmentations. Top-1 linear evaluation accuracies (%) and few-shot classification accuracy (%) are reported. All compared methods with a ResNet-50 encoder are pretrained for 200 epochs on the ImageNet-1k dataset.

| Method | Linear evaluation | | 5-way 1-shot | | 5-way 5-shot | |
|--------|-------|------------|-------|-------|-------|-------|
| | ImageNet | Caltech101 | FC100 | Plant | FC100 | Plant |
| CLSA [39] | 72.2 | 91.21 | 34.46 | 67.39 | 49.16 | 86.15 |
| RényiCL [40] | 72.6 | 94.04 | 36.31 | 80.68 | 53.39 | 94.29 |
| **CLVS** | **72.9** | **94.60** | **53.33** | **81.04** | **73.14** | **95.17** |

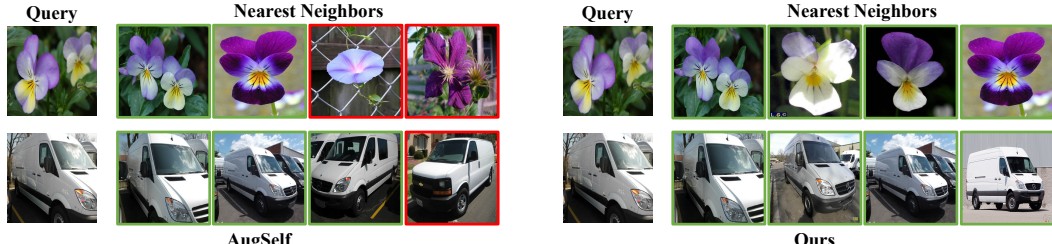

Figure 4: Visualization of retrieval results on the flowers and cars datasets. Red boxes indicate incorrect retrieval results, and green boxes indicate correct retrieval results.

Top-1 accuracy of 86.0% and 71.4% on the ImageNet-100 and ImageNet-1k datasets, significantly surpassing state-of-the-art methods. We also provide a comparison of training time among CLVS and other methods in Appendix D.

We further evaluate the performance of CLVS in transfer learning. We utilize the encoder pretrained on the ImageNet-100 dataset and conduct linear evaluation on 7 datasets: CIFAR10/100 [47], Caltech101 [48], SUN397 [49], Food [50], Flowers [51], and Pets [52]. The experimental results are presented in Tab. 3. CLVS achieves the best performance across all datasets, demonstrating its strong generalization ability and robustness.

**Comparison on Object Detection.** We also evaluate our method on the object detection task. The encoder pretrained on the ImageNet-100 dataset is converted to a generalized R-CNN detector following [11] and fine-tuned on the PASCAL VOC [53] and COCO [54] datasets. As shown in Tab. 4, CLVS demonstrates competitive performance on both datasets. Although CLVS performs slightly worse than CoCor on the smaller VOC dataset, where CoCor's strict similarity ordering is particularly effective in regularizing feature learning, it demonstrates a clear advantage on the larger and more complex COCO dataset. The performance improvement suggests that our proposed variable similarity can learn more generalizable feature representations.

**Comparison on Few-shot Classification.** In this experiment, we evaluate the performance of different contrastive learning methods on the few-shot classification task. We conduct experiments on FC100 [55], CUB200 [56], and Plant [57] datasets. Following the experimental setting of [35], we freeze the encoder pretrained on the ImageNet-100 dataset and employ logistic regression for classification. Tab. 5 presents the results of different methods under the 5-way 1-shot and 5-way 5-shot settings. In particular, the CUB and Plant datasets contain fine-grained labels, requiring models to have a strong generalizability to discern subtle differences. CLVS achieves excellent performance across all datasets, which can be attributed to the variable similarity mechanism that enables the model to capture fine-grained features, thereby significantly improving its generalization ability.

**Comparison on Retrieval.** We validate the effectiveness of our proposed method through visualization experiments. The image features are extracted by the encoder pretrained on the ImageNet-100 dataset, and the top-4 nearest neighbors are retrieved based on the cosine similarity metric. Fig. 4 presents the image retrieval results on the Flowers [51] and Cars [58] datasets. The retrieved images of flowers exhibit a smooth transition in color, starting with light-colored flowers and gradually transitioning to darker shades. Similarly, the retrieved results of cars show a progressive variation in object structure, ranging from close-up views to distant perspectives. These reflect the ability of our variable similarity framework to capture fine-grained visual attributes, ensuring that the retrieval results are not only semantically relevant but also visually coherent.

| $\mathcal{L}_{con}$ | $\mathcal{L}_d$ | $\mathcal{L}_{ali}$ | $\mathcal{L}_{re}$ | IN-100 | CIFAR10 | CIFAR100 |
|---|---|---|---|---|---|---|
| ✓ | | | | 79.1 | 86.5 | 66.0 |
| ✓ | ✓ | | | 80.5 (+0.6) | 88.1 (+1.6) | 68.7 (+2.7) |
| ✓ | ✓ | ✓ | | 85.5 (+5.4) | 89.8 (+3.3) | 71.1 (+5.1) |
| ✓ | ✓ | | ✓ | 85.0 (+4.9) | 89.4 (+2.9) | 70.8 (+4.8) |
| ✓ | ✓ | ✓ | ✓ | **86.0** (+5.9) | **89.9** (+3.4) | **71.4** (+5.4) |

Table 7: Ablation studies for different loss terms of CLVS. All results are reported on linear evaluation accuracy (%).

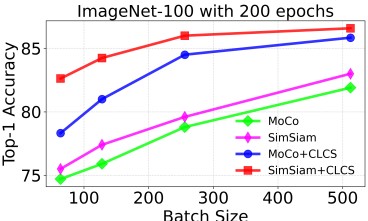

Figure 5: Top-1 accuracies of linear evaluation (%) with different batch sizes.

Table 8: Top-1 accuracies of linear evaluation (%) on various baselines. All methods with a ResNet-50 encoder are pretrained for 200 epochs on the ImageNet-100 dataset.

| Method | ImageNet-100 | CIFAR10 | CIFAR100 | Caltech101 | SUN397 | Food | Flowers | Pets |
|---|---|---|---|---|---|---|---|---|
| SimCLR [10] | 80.30 | 85.89 | 64.31 | 86.82 | 48.59 | 61.95 | 83.59 | 68.36 |
| **SimCLR + CLVS** | **83.48** (+3.18) | **88.15** (+2.26) | **68.45** (+4.14) | **89.18** (+2.36) | **52.63** (+4.04) | **66.13** (+4.18) | **88.03** (+4.44) | **73.15** (+4.79) |
| BYOL [13] | 83.24 | 87.32 | 66.95 | 87.27 | 49.97 | 63.89 | 85.31 | 75.01 |
| **BYOL + CLVS** | **85.80** (+2.56) | **89.86** (+2.54) | **71.27** (+4.32) | **89.92** (+2.65) | **53.42** (+3.45) | **67.83** (+3.94) | **89.88** (+4.57) | **77.62** (+2.61) |
| SwAV [26] | 75.48 | 83.87 | 60.70 | 84.52 | 44.81 | 58.39 | 79.83 | 62.58 |
| **SwAV + CLVS** | **81.08** (+5.60) | **86.84** (+2.97) | **66.76** (+6.06) | **88.42** (+3.90) | **51.14** (+6.33) | **64.13** (+5.74) | **87.27** (+7.44) | **68.96** (+6.38) |

**Comparison with Methods Using Stronger Augmentations.** To validate the capability of our proposed method in handling stronger augmentations, comparative experiments are conducted with several methods that employ stronger augmentations. Specifically, the proposed method is implemented using RényiCL as the baseline, with the encoder pretrained on the ImageNet-1k dataset. We perform the linear evaluation on the ImageNet-1k and Caltech101 datasets, and few-shot classification on FC100 and Plant datasets. The experimental results shown in Tab. 6 demonstrate that the proposed variable similarity can effectively mitigate the semantic inconsistency caused by stronger augmentations, resulting in a significant boost in linear evaluation performance.

## 4.3 Ablation Studies

In this subsection, we conduct ablation studies to evaluate the robustness of our method. All encoders are pretrained for 200 epochs on the ImageNet-100 dataset.

**Effect of Training Loss.** To evaluate the improvement contribution of each loss term, we conduct ablation studies on the ImageNet-100 dataset. The experimental results are shown in Tab. 7. First, we introduce a prediction loss based on the difference in augmentation parameters, which encourages the model to perceive augmentations, which yields a slight improvement. Second, the alignment loss $\mathcal{L}_{ali}$ drives the model to adjust embedding similarities in accordance with the semantic differences induced by data augmentations. This leads to a substantial performance gain, as it directly aligns feature similarities with semantic similarity. Third, the consistency loss $\mathcal{L}_{re}$ constrains the similarity between strongly and weakly augmented views. This further enhances the model's representation consistency under diverse augmentations. Combining all these loss terms leads to the best performance, demonstrating their complementary effects and validating the effectiveness of our proposed method. We also conduct experiments to validate the effectiveness of the augmentation-aware module, which can be found in Appendix D.

**Compatibility with Other Baselines.** We further integrate our proposed method into additional baseline models, including SimCLR [10], BYOL [13], and SwAV [26]. These methods are pretrained on ImageNet-100 and perform linear evaluation on the datasets mentioned in subsection 4.2. The experimental results in Tab. 8 demonstrate that our method consistently improves performance across all baseline models.

**Effect of Batch Size.** Given that contrastive learning is typically sensitive to batch size [10], we further investigate the impact of batch size on the performance of our proposed method. The experimental results are shown in Fig. 5. When the batch size increases, the accuracy of linear evaluation also improves. Even with a small batch size, our method maintains strong performance, demonstrating its good robustness.

# 5 Conclusion

In this paper, we introduced a variable similarity mechanism to extend the contrastive learning framework, which dynamically adjusts the similarity between views based on the extent of data augmentation. The variable similarity is estimated as the similarity between the refined features encoded by the augmentation-aware module, so that the variations induced by augmentations can be fully considered. By introducing an alignment loss for similarity and a relative similarity constraint between strong and weak augmentations, our method successfully addresses the semantic inconsistency caused by data augmentations. We also provided a theoretical analysis demonstrating that the variable similarity enhances the generalization error bound, validating its effectiveness. Experimental results indicated that the variable similarity enables the model to learn more robust feature representations. However, our method is limited to positive samples and does not incorporate negative samples into the variable similarity mechanism. In the future, we plan to extend the variable similarity to all samples to further optimize the contrastive learning framework.

## Acknowledgements

This work was supported by National Natural Science Fund of China (Nos. U24A20330, 62361166670 and 62506155), and Provincial Natural Science Fund of Jiangsu (Nos. BK20251985).

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

# A Proof

In this section, we present the proof for theory in the paper. We first introduce the following lemma for proving Theorem 1.

**Lemma 1.** *For independent random variables $t_1, t_2, \ldots, t_n \in \mathcal{T}$ and a given function $\omega : \mathcal{T}^n \to \mathbb{R}$, if $\forall v_i' \in \mathcal{T}$ ($i = 1, 2, \ldots, n$), the function satisfies*

$$|\omega(t_1, \ldots, t_i, \ldots, t_n) - \omega(t_1, \ldots, t_i', \ldots, t_n)| \leq \rho_i, \tag{9}$$

*then for any given $\mu > 0$, it holds that $P\{|\omega(t_1, \ldots, t_n) - \mathbb{E}[\omega(t_1, \ldots, t_n)]| > \mu\} \leq 2\mathrm{e}^{-2\mu^2 / \sum_{i=1}^{n} \rho_i^2}$.*

*Proof.* We prove Theorem 1 by analyzing the perturbation (i.e., $\rho_i$ in the above Eq. (9)) of the loss function $\mathcal{L}_{\mathrm{emp}}$.

We denote that

$$\omega = \mathcal{L}_{\mathrm{emp}}(\boldsymbol{f}, \boldsymbol{\varphi}; \mathscr{X}) = \frac{1}{N} \sum_{i=1}^{N} -\log \frac{\mathrm{e}^{sim(\boldsymbol{z}_1, \boldsymbol{z}_2)/\gamma}}{\sum_{j=1}^{n} \mathrm{e}^{sim(\boldsymbol{z}_1, \boldsymbol{z}_j)/\gamma}}, \tag{10}$$

and

$$\widetilde{\omega_r} = \frac{1}{N} \left[ \left( \sum_{i \neq r}^{N} -\log \frac{\mathrm{e}^{sim(\boldsymbol{z}_1, \boldsymbol{z}_2)/\gamma}}{\sum_{j=1}^{n} \mathrm{e}^{sim(\boldsymbol{z}_1, \boldsymbol{z}_j)/\gamma}}, \right) - \log \frac{\mathrm{e}^{sim(\widehat{\boldsymbol{z}}_1, \widehat{\boldsymbol{z}}_2)/\gamma}}{\sum_{j=1}^{n} \mathrm{e}^{sim(\widehat{\boldsymbol{z}}_1, \widehat{\boldsymbol{z}}_j)/\gamma}} \right], \tag{11}$$

where $(\widehat{\boldsymbol{x}}, \{\widehat{\boldsymbol{x}}_{b_j}\}_{j=1}^{n})$ is an arbitrary mini-batch from the sample space. Then we have that

$$|\omega - \widetilde{\omega_r}|$$

$$= \frac{1}{N} \left| \log \frac{\mathrm{e}^{sim(\widehat{\boldsymbol{z}}_1, \widehat{\boldsymbol{z}}_2)/\gamma}}{\sum_{j=1}^{n} \mathrm{e}^{sim(\widehat{\boldsymbol{z}}_1, \widehat{\boldsymbol{z}}_j)/\gamma}} - \log \frac{\mathrm{e}^{sim(\boldsymbol{z}_1, \boldsymbol{z}_2)/\gamma}}{\sum_{j=1}^{n} \mathrm{e}^{sim(\boldsymbol{z}_1, \boldsymbol{z}_j)/\gamma}} \right|$$

$$\leq \frac{1}{N} \log \left[ \frac{\mathrm{e}^{sim(\widehat{\boldsymbol{z}}_1, \widehat{\boldsymbol{z}}_2)/\gamma} \left( \mathrm{e}^{sim(\boldsymbol{z}_1, \boldsymbol{z}_2)/\gamma} + \sum_{j=1}^{n} \mathrm{e}^{sim(\widehat{\boldsymbol{z}}_1, \widehat{\boldsymbol{z}}_j)/\gamma} \right)}{\mathrm{e}^{sim(\boldsymbol{z}_1, \boldsymbol{z}_2)/\gamma} \left( \sum_{j=1}^{n} \mathrm{e}^{sim(\widehat{\boldsymbol{z}}_1, \widehat{\boldsymbol{z}}_j)/\gamma} + \mathrm{e}^{sim(\widehat{\boldsymbol{z}}_1, \widehat{\boldsymbol{z}}_2)/\gamma} \right)} \right]$$

$$\leq \frac{(C_1/(C_2 + \omega)) \log(1 + \sum_{i<j}(s_{ij} - \overline{s})/\mathrm{C}_N^2)}{2N}, \tag{12}$$

where $C_1, C_2 > 0$ are constants. Meanwhile, we have

$$\frac{1}{N} \sum_{i=1}^{N} -\log \frac{\mathrm{e}^{sim(\boldsymbol{z}_1, \boldsymbol{z}_2)/\gamma}}{\sum_{j=1}^{n} \mathrm{e}^{sim(\boldsymbol{z}_1, \boldsymbol{z}_j)/\gamma}} - \mathbb{E} \left( -\log \frac{\mathrm{e}^{sim(\boldsymbol{z}_1, \boldsymbol{z}_2)/\gamma}}{\sum_{j=1}^{n} \mathrm{e}^{sim(\boldsymbol{z}_1, \boldsymbol{z}_j)/\gamma}} \right)$$

$$= \mathcal{L}_{\mathrm{emp}}(\boldsymbol{f}, \boldsymbol{\varphi}; \mathscr{X}) - \widetilde{\mathcal{L}}_{\mathrm{emp}}(\boldsymbol{f}, \boldsymbol{\varphi}; \mathscr{D}). \tag{13}$$

By Lemma 1, we let that for all $i = 1, 2, \ldots, N$

$$\rho_i = \frac{(C_1/(C_2 + \omega)) \log(1 + \sum_{i<j}(s_{ij} - \overline{s})/\mathrm{C}_N^2)}{2N}, \tag{14}$$

so that we have

$$P \left\{ \left| \mathcal{L}_{\mathrm{emp}}(\boldsymbol{f}, \boldsymbol{\varphi}; \mathscr{X}) - \widetilde{\mathcal{L}}_{\mathrm{emp}}(\boldsymbol{f}, \boldsymbol{\varphi}; \mathscr{D}) \right| < (C_1/(C_2 + \omega)) \log(1 + \sum_{i<j}(s_{ij} - \overline{s})/\mathrm{C}_N^2) \sqrt{\frac{\ln(2/\delta)}{2N}} \right\}$$

$$= 1 - 2\mathrm{e}^{-2\mu^2 / \sum_{i=1}^{N} \rho_i^2}$$

$$\geq 1 - 2\mathrm{e}^{\frac{-2N(\eta\sqrt{[\ln(2/\delta)]/(2C\lambda N)})^2}{(C_1/(C_2+\omega))\log(1+\sum_{i<j}(s_{ij}-\overline{s})/\mathrm{C}_N^2)}}$$

$$= 1 - 2\mathrm{e}^{-2N\left(\sqrt{[\ln(2/\delta)]/(2C\lambda N)}\right)^2}$$

$$= 1 - 2\mathrm{e}^{-\ln(2/\delta)}$$

$$= 1 - \delta, \tag{15}$$

where $\eta = \frac{(C_1/(C_2+\omega))\log(1+\sum_{i<j}(s_{ij}-\overline{s})/\mathrm{C}_N^2)}{2N}$ and $\mu = \sqrt{[\ln(2/\delta)]/(2N)}$. The proof is completed.
$\square$

# B  Pretraining Setup and Evaluation Protocols

## B.1  Pretraining Setup

In the experiments, we integrate CLVS into MoCo and SimSiam. Here, we provide the detailed pretraining setup used for each method:

- MoCo. The learning rate is set to 0.03 and the temperature parameter for contrastive loss is 0.2. The projector consists of 2 MLP layers with an output dimension of 128. The memory queue size is 65536 and the exponential moving average (EMA) parameter is 0.999.
- SimSiam. The learning rate is set to 0.05. The projector and predictor consist of 3 MLP layers and 2 MLP layers with an output dimension of 2048, respectively.

## B.2  Linear Evaluation

The linear evaluation protocol on ImageNet-100 and ImageNet-1k datasets follows [12]. Specifically, we freeze the backbone and train a classifier by minimizing the cross-entropy loss using the LARS [59] optimizer.

The linear evaluation protocol for transfer learning follows [35]. The training datasets are split into a train set and a validation set, with 90% for training and the remaining 10% for validation. The representation of $224 \times 224$ center-cropped images are extracted by the frozen backbone. The classifier is trained by minimizing the $L_2$-regularized cross-entropy loss using a L-BFGS [60] optimizer.

## B.3  Object Detection

The object detection protocol follows [11]. We train the Faster R-CNN [61] detector with a backbone of ResNet-50-C4 [62]. The detector is fine-tuned end-to-end on all layers for $24000$ iterations on the VOC dataset and $180000$ iterations on the COCO dataset. During training, the image scale is resized to $[480, 800]$ pixels for VOC and $[640, 800]$ pixels for COCO, while a fixed scale of $800$ pixels is used during inference.

## B.4  Few-Shot Classification

The few-shot classification protocol follows [35]. We conduct logistic regression using representation extracted by the frozen backbone from $224 \times 224$ images in an $N$-way $K$-shot episode.

# C  More Discussion on Related Work

Our method and Augself [35] both argue that the augmentation-invariance assumption in traditional contrastive learning may harm semantic consistency. However, the two approaches address this issue from completely different perspectives. To be specific, AugSelf uses an auxiliary task to only preserve augmentation information in its forward phase. In contrast, our method further explores the augmentation information to explicitly modulate the variable contrastive-supervision signal for both forward and backward phases, leading to three important technical differences:

**Dynamic Similarity Adjustment.** We introduce a variable similarity mechanism that dynamically adjusts the similarity supervision between two augmented views. Nevertheless, AugSelf does not incorporate such a dynamic mechanism, and thus it cannot explicitly adjust the similarity supervision in the corresponding learning objective.

**Relative Similarity Constraint.** We further introduce a relative constraint that enforces the similarity between weakly augmented views to be reasonably higher than that between weakly and strongly augmented views. This relational structure ensures the important semantic consistency, which is absent in AugSelf.

**Augmentation-Type Aware Similarity Estimation.** AugSelf treats all augmentation types equally. In contrast, our method explicitly accounts for the transformation that induces the most significant semantic change and uses the minimum similarity across types as the target. This practice allows the model to focus on the most challenging transformation, leading to robust representation.

# D   Additional Experimental Results

## D.1   Effect of Augmentation-Aware Module

We conduct experiments with the augmentation-aware module $\varphi$. The experimental results in Tab. 9 demonstrate the effectiveness of $\varphi$, which benefits the representation robustness via reasonably predicting the similarity between views caused by different augmentations.

## D.2   Effect of Minimum Similarity Strategy

We conduct experiments with different strategies of similarity selection in variable similarity. The experimental results in Tab. 10 demonstrate the effectiveness of the minimum similarity strategy. These results support our hypothesis that focusing on the most challenging cases enhances representation robustness.

Table 9: Effect of the augmentation-aware module in CLVS to the linear evaluation accuracy (%). All methods are pretrained for 200 epochs on the ImageNet-100 dataset.

| Method | Baseline | w/o $\varphi$ | w $\varphi$ |
|---|---|---|---|
| MoCo+CLVS | 78.8 | 83.1 | **84.5** |
| SimSiam+CLVS | 79.1 | 84.5 | **86.0** |

Table 10: Top-1 accuracies of linear evaluation (%) with different strategies. All methods with a ResNet-50 encoder are pretrained for 200 epochs on the ImageNet-100 dataset.

| Method | Min | Max | Mean |
|---|---|---|---|
| MoCo+CLVS | **84.5** | 84.0 | 83.8 |
| SimSiam+CLVS | **86.0** | 85.4 | 85.7 |

## D.3   More Epochs

In the aforementioned experiments, we adopt the setup of 200 training epochs from prior work [41]. To comprehensively evaluate model performance at convergence, we conduct experiments with more training epochs. Specifically, we pretrain both CLVS and comparative methods on the ImageNet-100 dataset for 800 epochs and perform linear evaluation. As shown in Tab. 11, our method consistently outperforms comparative methods. These extended results further corroborate and align with the findings presented in Table 3, reinforcing the robustness and superiority of our proposed method.

Table 11: Top-1 accuracies of linear evaluation (%). All methods with a ResNet-50 encoder are pretrained for 800 epochs on the ImageNet-100 dataset.

| Method | ImageNet-100 | CIFAR10 | CIFAR100 | Caltech101 | SUN397 | Food | Flowers | Pets |
|---|---|---|---|---|---|---|---|---|
| MoCo [16] | 85.7 | 87.5 | 67.0 | 89.1 | 51.3 | 62.8 | 84.9 | 76.6 |
| SimSiam [12] | 81.6 | 88.1 | 68.7 | 89.7 | 52.1 | 64.6 | 88.0 | 75.3 |
| AugSelf [35] | 83.3 | 89.7 | 71.7 | 91.4 | 53.2 | 68.3 | 90.5 | 77.0 |
| RényiCL [40] | 86.6 | 90.0 | 69.8 | 90.5 | 55.4 | 65.6 | 89.9 | 77.9 |
| EquiMod [38] | 86.2 | 88.6 | 70.0 | 89.6 | 54.0 | 66.7 | 89.3 | **79.7** |
| GCA [44] | 74.0 | 75.2 | 49.7 | 78.0 | 37.3 | 43.6 | 70.6 | 52.2 |
| INTL [32] | 86.5 | 88.3 | 69.5 | 89.9 | 53.6 | 64.5 | 89.0 | 79.0 |
| **MoCo+CLVS** | **88.7** | 88.4 | 68.3 | 90.6 | 54.9 | 66.7 | 88.9 | 78.7 |
| **SimSiam+CLVS** | 87.6 | **91.0** | **73.4** | **92.1** | **56.6** | **69.6** | **91.3** | 79.2 |

## D.4   ViT Backbone

To further validate the effectiveness of our proposed method, we conduct experiments using the ViT [63] backbone. Recently, MoCo-v3 [64] explores training ViT backbone in self-supervised learning framework. Therefore, we implement our method into MoCo-v3. We pretrain the ViT-Small encoder on the Imagenet-100 dataset for 200 epochs with the batch size of 1024. We follow the experimental settings in MoCo-v3, including AdamW optimizer [65] with a linear learning rate warm-up for the first 40 epochs, a momentum of 0.9, and a weight decay of 0.1. A cosine learning rate schedule is applied to the encoder and predictor. The learning rate is set to 1.5e-4 and temperature is 0.2. The experimental results are shown in Tab. 12. The proposed method with the ViT backbone exhibits better performance across all datasets. This reveals the effectiveness of the variable similarity, which is applicable of the ViT backbone.

Table 12: Top-1 accuracies of linear evaluation (%). All methods with a ViT-Small encoder are pretrained for 200 epochs on the ImageNet-100 dataset.

| Method | ImageNet-100 | CIFAR10 | CIFAR100 | Caltech101 | SUN397 | Food | Flowers | Pets |
|---|---|---|---|---|---|---|---|---|
| MoCo-v3 [16] | 78.4 | 86.2 | 66.4 | 82.2 | 47.1 | 60.5 | 83.7 | 66.9 |
| **MoCo-v3 + CLVS** | **80.7** | **86.9** | **68.5** | **85.3** | **49.5** | **63.9** | **86.8** | **69.7** |

## D.5 Impact of Individual Data Augmentations

To better understand how different data augmentations contribute to variable similarity estimation, we perform an ablation where each augmentation type is applied in isolation within the SimSiam framework. Specifically, we consider Random Cropping, Color Jittering, Gaussian Blurring, and Horizontal Flipping, and evaluate the similarity prediction using only the parameter difference from the chosen augmentation. As shown in Table 13, cropping and color jittering yield the strongest performance, indicating that view differences along spatial and color dimensions provide more informative cues for similarity estimation. In contrast, blurring and flipping lead to slightly lower accuracy, likely because they have limited impact on semantic content: flipping only changes orientation and blurring mainly suppresses low-level details. This suggests that augmentations that alter semantically relevant aspects of the image are more effective for guiding similarity estimation.

Table 13: Linear evaluation accuracy (%) with individual augmentations. The method uses a ResNet-50 encoder pretrained for 200 epochs on ImageNet-100, with only one augmentation type applied during similarity estimation.

| Method | Random Cropping | Color Jittering | Gaussian Blurring | Horizontal Flipping |
|---|---|---|---|---|
| **SimSiam + CLVS** | 85.1 | 84.9 | 84.4 | 84.2 |

## D.6 Parametric Sensitivity

We systematically analyze the sensitivity of the loss hyperparameters $\omega$, $\lambda$ and $\gamma$ by varying each in $0.1, 0.5, 1$. As shown in Table 14, $\omega$ exhibits the clearest trend, i.e., intermediate values consistently lead to the best results, while both small and large values reduce accuracy. The effect of $\lambda$ is similar, where performance peaks at 0.5 and slightly declines at 1. In contrast, $\gamma$ shows relatively stable behavior across different values, with a marginal gain at $\gamma = 1$, indicating lower sensitivity. Overall, these findings suggest that CLVS is robust to hyperparameter variations within a reasonable range, with optimal performance typically achieved at moderate settings.

Table 14: Parametric sensitivity of different hyperparameters in the training loss. Linear evaluation accuracy (%) of ResNet-50 backbone encoder pretrained for 200 epochs on the ImageNet-100 dataset.

| Parameter | Value | MoCo+CLVS | SimSiam+CLVS |
|---|---|---|---|
| | 0.1 | 84.1 | 85.4 |
| $\omega$ | 0.5 | **84.9** | **86.0** |
| | 1 | 83.9 | 85.1 |
| | 0.1 | 84.4 | 85.8 |
| $\lambda$ | 0.5 | **84.9** | **86.0** |
| | 1 | 84.5 | 85.2 |
| | 0.1 | 84.1 | 85.8 |
| $\gamma$ | 0.5 | 84.7 | 85.8 |
| | 1 | **84.9** | **86.0** |

## D.7 Training Time Comparison

We provide experiments to record the training time of our method and the compared methods. Specifically, we use 4 NVIDIA A100-SXM4-40GB GPUs to train these methods, where the batch

size is set to 256. As shown in Tab. 15, although our method introduces an extra module $\varphi$, it only increases the time by around $5\%$ compared to the baseline. This demonstrates competitive efficiency in terms of training time.

Table 15: Time comparison of our proposed method and compared methods on the ImageNet-1k dataset (in minutes).

| Method | Time / 1 epoch |
|---|---|
| SimSiam [12] | 24.22 |
| RényiCL [40] | 24.72 |
| EquiMod [38] | 38.83 |
| INTL [32] | 29.17 |
| CLVS | 25.78 |

Table 16: NMI score of our methods and compared method to quantify the class separation on the ImageNet-100 dataset.

| Method | NMI |
|---|---|
| SimSiam | 0.69 |
| Augself | 0.74 |
| RényiCL | 0.68 |
| CLVS | **0.76** |

## D.8 Visualization of Learned Representations

We show the t-SNE [66] visualizations of the representations learned by our proposed method and several methods on the ImageNet-100 dataset. As shown in Fig. 6, our proposed method leads to better class separation. This indicates that variable similarity facilitates the model to learn more discriminative feature representations. Furthermore, we also provide the NMI (Normalized Mutual Information) score to quantify the class separation in Tab. 16. Our method achieves the highest NMI score, aligning with the trends observed in t-SNE visualizations.

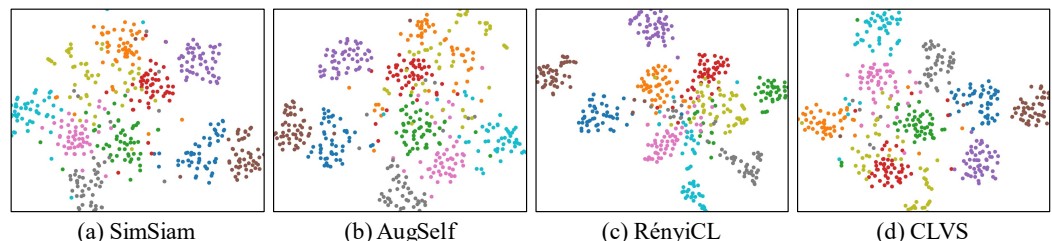

    (a) SimSiam          (b) AugSelf          (c) RényiCL          (d) CLVS

Figure 6: t-SNE visualizations of image representations from randomly selected 10 classes in the ImageNet-100 validation set.

