# OpenReview forum: "Enhancing Contrastive Learning with Variable Similarity"
_NeurIPS.cc/2025/Conference — NeurIPS 2025 spotlight_

### Official Review · Reviewer_LeRi · 2025-06-27

**Clarity:** 3
**Significance:** 3
**Originality:** 3
**Rating:** 5
**Confidence:** 5

**Summary:**

This paper proposes a novel contrastive learning method with variable similarity. The authors argue that different augmentations introduce varying degrees of semantic bias, and therefore, positive samples should not be treated as uniformly similar. They propose a variable similarity objective to dynamically adjust the similarity between positive pairs.

The key idea is intuitive, which builds an augmentation-aware module to predict the variable similarity based on augmentation parameters. The proposed similarity alignment loss and relative constraint loss enable the model to more accurately reflect the true semantic distance between augmented views, promoting better representation learning.

The paper supports this idea with both theoretical and empirical evidence. The authors provide a generalization error bound analysis to justify the improved learning behavior under variable similarity conditions. Nowadays, it is unusual to see some theoretical finds in deep learning and self-supervised learning. Empirically, they evaluate this method across multiple downstream tasks, demonstrating consistent robustness.

**Questions:**

Please refer to Weaknesses.

**Ethical Concerns:**

["NO or VERY MINOR ethics concerns only"]

**Final Justification:**

My concerns have been addressed, so I will maintain my positive rating.

**Limitations:**

yes

**Paper Formatting Concerns:**

I have not noticed any major formatting issues.

**Quality:**

3

**Strengths And Weaknesses:**

**Strengths**

1.The motivation of this paper is clear and interesting, which points out the semantic similarity between positive samples varies depending on data augmentations. The proposed method comprises of various loss terms - they are intuitive, make sense and well formulated.

2.The experimental results across multiple datasets demonstrate consistent performance gains, reflecting the robustness and generalizability of the proposed method. The validation of its general applicability highlights the method’s potential for broader use.


3.This paper provides a clear and well-reasoned theoretical analysis. I appreciate the theoretical evidence of the method, as it enhances the rigor of the paper.

4.This paper is well written, with a clear and logic structure. It is easy to follow the problem that the authors claimed and their solution.


**Weaknesses**

There are some minor weaknesses in my opinion.

1.It seems that CoCor also defines a similar notion of augmentation strength. What distinguishes this paper from CoCor? This has important implications for the novelty of this work.

2.Although the method aggregates the minimum similarity across different augmentation types, it is not clear how individual augmentation operations affect the predicted similarity. The authors could conduct experiments on this.

---

> ### Author Rebuttal · Authors · 2025-07-30
>
> We appreciate the reviewer's positive comments for our paper. We address the concerns raised point by point below.
>
> ---
>
> **Comment_1:** Differences of proposed method with CoCor.
>
> **Response_1:** Thanks for your comment. We would like to clarify that CoCor and our proposed method differ significantly in both methodology and design philosophy.
>
> CoCor defines augmentation strength based on the frequency or number of transformations applied to a sample. Specifically, a composite augmentation with more operators is defined as a stronger augmentation. Based on this, CoCor proposed a data augmentation consistency, i.e., the similarity between the augmented view and the original input decreases with the strength of the augmentation. However, the learning of this similarity structure in CoCor is supervised by ground-truth class labels, which are required to guide the alignment of augmented views with their semantic categories. This makes the approach label-dependent and less applicable in fully unsupervised settings.
>
> In contrast, our method is based on a different perspective of augmentation. Even though positive pairs are derived from the same data sample, different parameterizations within the same augmentation type will induce variations in the resulting views. Therefore, the similarity between such augmented views should not be assumed constant, but rather should vary in accordance with the applied augmentation parameters. We define similarity not based on the number of transformations, but rather based on the parameter variations within the same augmentation type. For instance, two crops with small scale differences are considered more similar than two crops with large scale difference. Our method predicts this similarity via an augmentation-aware module $ \varphi $ and aligns it with the representation similarity in a self-supervised manner. This enables the model to capture semantic consistency from augmentation parameters, without relying on human-annotated labels.
>
> In summary, while both approaches utilize augmentation-related cues, our method is fundamentally different in its definition of similarity, learning strategy, and reliance on supervision. We believe these differences not only constitute a clear methodological novelty, but also broaden the applicability of our approach to a wider range of unsupervised learning scenarios.
>
> ---
>
> **Comment_2:** Effectiveness of individual augmentation.
>
> **Response_2:** Thanks for your comment. To better understand the effect of individual augmentation types on the predicted similarity, we conducted an ablation study to evaluate each augmentation’s contribution.
>
> Specifically, we conduct this experiment based on the SimSiam method. We isolate the effect of each of the following augmentations: random cropping, color jittering, Gaussian blurring, and horizontal flipping. When predicting the similarity between two augmented views, we use only the parameter difference from a single augmentation type, effectively assessing its individual contribution to the variable similarity estimation. The following results are obtained:
>
> | Type | Cropping | Color Jittering | Gaussian Blurring | Flipping |
> | :-: | :-: | :-: | :-: | :-: |
> | Top-1 | 85.1 | 84.9 | 84.4 | 84.2 |
>
> These results reveal several insights: Crop and color jitter yield stronger performance when used alone, suggesting that their parameter differences offer more informative cues about view similarity. In contrast, blurring and flipping result in slightly lower performance. This may be attributed to their limited influence on the image’s semantic content, as flipping alters only the orientation and blurring affects low-level details without substantially changing object-level features. Consequently, their limited semantic impact makes them less effective in providing informative variation for similarity estimation.
>
> Thanks a lot again for your positive comments on our paper! We sincerely hope that our responses can further improve the clarity of the paper.

---

> > ### Comment · Reviewer_LeRi · 2025-08-06
> > **Response**
> >
> > Thank you for the response. My concerns have been addressed, so I will maintain my positive rating.

---

> > > ### Author Response · Authors · 2025-08-06
> > > **Thank you a lot!**
> > >
> > > Dear Reviewer LeRi,
> > >
> > > Thank you very much for maintaining a positive rating on our paper.
> > >
> > > Best wishes,
> > >
> > > Authors of Paper ID-11483

---

### Official Review · Reviewer_HTKK · 2025-06-28

**Clarity:** 3
**Significance:** 3
**Originality:** 4
**Rating:** 5
**Confidence:** 5

**Summary:**

This paper introduces Contrastive Learning with Variable Similarity (CLVS), a novel extension to existing contrastive learning frameworks that dynamically adjusts the similarity objective between positive pairs based on the strength of their data augmentations.
By incorporating an augmentation-aware module to predict semantic consistency and enforcing both an alignment loss (to match predicted similarity) and a relative similarity constraint (to ensure stronger augmentations yield lower similarity), the authors demonstrate significant improvements over state-of-the-art baselines on multiple benchmarks.
In particular, CLVS achieves up to +6.1% gain on ImageNet-100 with MoCo and +5.9% with SimSiam, as well as consistent improvements across transfer learning, detection, few-shot classification, and retrieval tasks.

CLVS addresses a meaningful limitation in contrastive learning with strong theoretical and empirical evidence, and its general-purpose design promises impact across a wide range of self-supervised tasks.

**Questions:**

See the weaknesses.

**Ethical Concerns:**

["NO or VERY MINOR ethics concerns only"]

**Limitations:**

yes

**Paper Formatting Concerns:**

I have not noticed major formatting issues at this moment.

**Quality:**

4

**Strengths And Weaknesses:**

I found the paper has several strong points:

$Strengths$

i). Clear Motivation & Novelty.

The observation that aggressive augmentations can introduce semantic distortion is well-motivated, and modeling similarity as a continuous variable rather than a binary constant is both intuitive and underexplored in prior work.

ii). Theoretical Support.

A rigorous generalization error bound analysis (Theorem 1) shows how the additional loss terms tighten the contrastive learning objective’s generalization gap, providing solid theoretical backing.

iii). Comprehensive Experiments.

Extensive evaluation on ImageNet-100/1k, seven transfer datasets, PASCAL VOC & COCO detection, three few-shot benchmarks, and retrieval tasks convincingly demonstrate CLVS’s broad applicability and robustness.

iv). Ablation & Compatibility.

Detailed ablations dissect the contributions of each loss component and the minimum-similarity strategy, while integration with various baselines (SimCLR, BYOL, SwAV) shows the method’s versatility.


I also want to list some questions and weaknesses here:

$Weaknesses$

i) Clarity of Notation (Sec. 3). Equations (4)–(6) introduce several similarity terms (e.g., s_1,2, s^_1,2) in quick succession. A consolidated notation table or a more gradual build-up may help readability.

ii) Parameter Sensitivity. While the paper sets weights ω, λ, γ empirically, a brief sensitivity analysis (e.g., grid search results) would help practitioners understand tuning requirements.

iii) Negative Sampling Extension. It seems that the current method focuses solely on positive pairs. A discussion or preliminary experiment on extending variable similarity to negative samples (e.g., weighting negatives by augmentation distance) would be a valuable direction.

---

> ### Author Rebuttal · Authors · 2025-07-30
>
> Thank you for your appreciation of the novelty, theoretical analyses, and experimental results of our paper! Thanks also for your very insightful and constructive suggestions! Our point-by-point responses are as follows.
>
> ---
>
> **Comment_1:** Clarity of Notation.
>
> **Response_1:** We appreciate the reviewer for pointing out this issue. To clarify, we summarize the meaning and role of each similarity term used:
>
> | Notation | Description |
> | :-: | :-: |
> | $\hat{s}_{1,2}^j$ | Predicted cosine similarity under $j$-th augmentation |
> | $\hat{s}_{1,2}$ | Minimum similarity across all augmentations |
> | $s_{1,2}$ | Cosine similarity of raw embeddings |
>
> We will introduce the above notations with clearer textual explanation accompanying each equation if the paper is finally accepted.
>
> ---
>
> **Comment_2:** Parameter Sensitivity.
>
> **Response_2:** Thanks for your comment. The sensitivity analyses of $ \lambda $ and $ \gamma $ have already been presented in Tab. 9 and 10 of our manuscript, respectively. The value of $ \omega $ follows the same setting as in AugSelf [R1]. To further address the reviewer’s concern, we additionally conduct a sensitivity analysis for $ \omega $, while fixing $ \lambda = 0.5 $ and $ \gamma = 1 $.
>
> | $ \omega $ | 0.1 | 0.5 | 1 |
> | :-: | :-: | :-: | :-: |
> | MoCo+CLVS | 84.1 | **84.9** | 83.9 |
> | SimSiam+CLVS | 85.4 | **86.0** | 85.1 |
>
> These results indicate that the performance is relatively stable across a reasonable range of $ \omega $, suggesting that the method is not highly sensitive to this parameter.
>
> ---
>
> **Comment_3:** Negative Sampling Extension.
>
> **Response_3:** Thanks for your comment.
>
> Our method focuses on adjusting the similarity objective for positive pairs, as they are directly generated from the same instance via data augmentations. In this case, the augmentation parameters provide a meaningful and grounded basis for modeling semantic similarity.
>
> Extending the concept of variable similarity to negative pairs is indeed an interesting direction. However, we note a fundamental difference: negative pairs typically consist of different samples, and their similarity arises not only from augmentations but also from inherent semantic differences. As such, using augmentation parameters alone to model the similarity of negative pairs may not be sufficient or reliable.
>
> One potential solution could be inspired by information-theoretic principles: we may estimate the inherent similarity between samples (e.g., using approximations of mutual information), and combine it with augmentation-induced variation to form a more comprehensive similarity measure for negative pairs. This would allow the model to jointly consider intrinsic sample differences and augmentation-induced discrepancies when weighting negative similarities.
>
> We believe this integration requires careful design and is beyond the scope of the current work, but we appreciate the reviewer’s suggestion and will consider it as a future research direction.
>
> Thanks a lot again for your very detailed comments on our paper! We sincerely hope that our responses can further improve the clarity of the paper.
>
> [R1] Improving Transferability of Representations via Augmentation-Aware Self-Supervision, NIPS, 2021.

---

> > ### Comment · Reviewer_HTKK · 2025-08-05
> >
> > Thank you for your detailed rebuttal, it solved my concerns and I will maintain my positive score on this paper.

---

> > > ### Author Response · Authors · 2025-08-06
> > > **Thank you a lot!**
> > >
> > > Dear Reviewer HTKK,
> > >
> > > Thank you very much for keeping positive on our paper.
> > >
> > > Best wishes,
> > >
> > > Authors of Paper ID-11483

---

### Official Review · Reviewer_rS1f · 2025-06-30

**Clarity:** 3
**Significance:** 3
**Originality:** 3
**Rating:** 5
**Confidence:** 4

**Summary:**

Different from most existing self-supervised learning (SSL) approaches, the authors argue that different augmentation methods can lead to semantic discrepancies. They use variable similarity to accurately characterize the intrinsic similarity relationships between different augmented views, thereby enhancing the performance of existing SSL approaches. In addition to empirical analyses, the authors also offer theoretical analysis to demonstrate the effectiveness of variable similarity.

**Questions:**

(1) I’m extremely curious about the similarities and differences between authors' paper and AugSelf [1]. As mentioned in **Weakness** (1), it appears that authors' paper and AugSelf share the same motivation. I strongly recommend that the authors could discuss the similarities and differences with AugSelf. This will enable readers to better understand your work. I’ll adjust my review score based on the discussion the authors provide.

(2) Most self-supervised learning (SSL) papers conduct experiments on ImageNet-1k and typically provide results for 800-1000 epochs. However, the authors only provided the linear evaluation results for the first 200 epochs, e.g., in Tables 1 and 2. I’m curious whether the effectiveness of the method proposed by the authors will diminish as the number of epochs increases.

[1] Improving Transferability of Representations via Augmentation-Aware Self-Supervision

**Ethical Concerns:**

["NO or VERY MINOR ethics concerns only"]

**Final Justification:**

I am writing to provide my justification for raising the score of this submission to 5.

Upon careful review and considering the rebuttal, I have come to the conclusion that this paper demonstrates a high level of novelty. The innovative ideas presented in the paper show great potential to contribute to the field.

One of my major concerns initially was the comparison with AugSelf. However, during the rebuttal process, the authors effectively addressed this issue. They provided detailed explanations and additional analyses that successfully clarified the differences and advantages of their approach over AugSelf. This resolution of the concern has significantly enhanced my confidence in the paper’s quality.

After thorough consideration of the overall content, the rebuttal, and the discussions with the authors, other reviewers, and you, I am convinced that this submission has reached the acceptance level. Therefore, I have decided to raise my score to 5.

**Limitations:**

yes

**Quality:**

3

**Strengths And Weaknesses:**

**Strengths:**

(1) The paper is strongly motivated. As the assumption that different augmentation methods can lead to semantic discrepancies is reasonable and interesting.

(2) The authors have provided numerous experiments to prove the effectiveness of their method.

(3) The authors not only provide empirical analyses on variable similarity but also offer theoretical analysis.


**Weakness:**

(1) While the authors’ motivation is interesting, the paper AugSelf [1] seems to be the first one to propose a solution to the semantic discrepancies arising from augmentation methods. This is my biggest concern. The authors seem not to have provided a comprehensive discussion on this aspect.

(2) Most self-supervised learning (SSL) papers conduct experiments on ImageNet-1k and typically provide results for 800-1000 epochs. However, the authors only provided the linear evaluation results for the first 200 epochs, e.g., in Tables 1 and 2.

[1] Improving Transferability of Representations via Augmentation-Aware Self-Supervision

---

> ### Author Rebuttal · Authors · 2025-07-30
>
> Thank you for your positive and constructive comments! Our point-by-point responses are provided below.
>
> ---
>
> **Comment_1:** Similarities and differences with AugSelf.
>
> **Response_1:** Thanks for your comment. As the reviewer suggested, our proposed method and AugSelf both argue that the augmentation-invariance assumption in traditional contrastive learning may harm semantic consistency. However, the two approaches address this issue from *completely different* perspectives.
>
> To be specific, AugSelf uses an auxiliary task to only *preserve* augmentation information in its forward phase. In contrast, our method further explores the augmentation information to explicitly *modulate* the variable contrastive-supervision signal for both forward and backward phases, leading to three important technical differences:
>
> 1. *Dynamic Similarity Adjustment*: We introduce a variable similarity mechanism that dynamically adjusts the similarity supervision between two augmented views. Nevertheless, AugSelf does not incorporate such a dynamic mechanism, and thus it cannot explicitly adjust the similarity supervision in the corresponding learning objective.
>
> 2. *Relative Similarity Constraint*: We further introduce a relative constraint that enforces the similarity between weakly augmented views to be reasonably higher than that between weakly and strongly augmented views. This relational structure ensures the important semantic consistency, which is absent in AugSelf.
>
> 3. *Augmentation-Type Aware Similarity Estimation*: AugSelf treats all augmentation types equally. In contrast, our method explicitly accounts for the transformation that induces the most significant semantic change and uses the minimum similarity across types as the target. This practice allows the model to focus on the most challenging transformation, leading to robust representation.
>
> We will add the above discussion in the camera-ready version if this paper is finally accepted.
>
> ---
>
> **Comment_2:** Effectiveness of more epochs.
>
> **Response_2:** Thanks for your comment.
>
> We would like to clarify that the number of training epochs in our experiments follows the same setting as in previous works [R1, R2, R3], where 200 epochs are commonly used for evaluations on both the ImageNet-100 and ImageNet-1k datasets. Moreover, the results of Tab. 11 in the appendix show that our proposed method remains effective when trained for 800 epochs on the ImageNet-100 dataset, indicating that its effectiveness is successfully demonstrated with more training epochs.
>
> For ImageNet-1k, here we follow the reviewer's suggestion to train our model with 800 epochs by using 8 NVIDIA Tesla V100 GPUs. The training setup follows the baseline method SimSiam, with a learning rate of 0.05, and batch size of 256. The corresponding results are shown in the following table.
>
> | Method | SimCLR | MoCo | SwAV | SimSiam | INTL | Ours |
> | :-: | :-: | :-: | :-: | :-: | :-: | :-: |
> | Top-1 | 70.4 | 71.1 | 71.8 | 71.3 | 73.1 | **73.7** |
>
> As shown in the above table, our method achieves the best top-1 accuracy on the ImageNet-1k dataset among all compared methods. The experimental results demonstrate the effectiveness of our proposed method with more training epochs.
>
> Thanks a lot again for your very insightful comments on our paper! We sincerely hope that our responses can solve your main concerns.
>
> [R1] Contrastive Learning with Consistent Representations, TMLR, 2024.
>
> [R2] Contrastive Learning with Stronger Augmentations, TPAMI, 2023.
>
> [R3] On the Effectiveness of Supervision in Asymmetric Non-Contrastive Learning, ICML, 2024.

---

> ### Comment · Reviewer_rS1f · 2025-08-01
> **Feedback to Authors**
>
> Thanks for the authors’ detailed responses. I think the authors have solved my concerns, so I will raise the score to 5.

---

> > ### Author Response · Authors · 2025-08-02
> > **Thanks a lot!**
> >
> > Dear Reviewer rS1f,
> >
> > Thank you so much for your great patience and your appreciation for our paper!
> >
> > Best wishes,
> >
> > Authors of Paper ID-11483

---

### Decision · Program_Chairs · 2025-09-17

**Decision:**

Accept (spotlight)

**Comment:**

This paper studies modifying the SSL by the strength of the augmentation between positive pairs, ensuring sensitivity to semantic biases found in the augmentation transformation. The approach appears to be useful across SSL algorithms / losses, and hence would have high use for practitioners. The reviewers had a few issues, notably with similarities to a few other works that build in sensitivity to augmentation parameters, but the authors well-distinguished their work for those, both in motivation and implementation.

I therefore recommend the paper is accepted as a spotlight.